# A Regulatory SRNA Rli43 Is Involved in the Modulation of Biofilm Formation and Virulence in *Listeria monocytogenes*

**DOI:** 10.3390/pathogens11101137

**Published:** 2022-09-30

**Authors:** Lixia Wang, Chunhui Ji, Xianzhu Xia, Xuepeng Cai, Qingling Meng, Jun Qiao

**Affiliations:** 1College of Animal Science and Technology, Shihezi University, Shihezi 832003, China; 2State Key Lab of Veterinary Etiological Biology, Lanzhou Veterinary Research Institute, Chinese Academy of Agricultural Sciences, Lanzhou 730046, China

**Keywords:** *Listeria monocytogenes*, Rli43, biofilm formation, virulence

## Abstract

Small RNAs (sRNAs) are a kind of regulatory molecule that can modulate gene expression at the post-transcriptional level, thereby involving alteration of the physiological characteristics of bacteria. However, the regulatory roles and mechanisms of most sRNAs remain unknown in *Listeria monocytogenes*
*(L. monocytogenes*). To explore the regulatory roles of sRNA Rli43 in *L. monocytogenes*, the *rli43* gene deletion strain LM-*Δrli43* and complementation strain LM-*Δrli43-rli43* were constructed to investigate the effects of Rli43 on responses to environmental stress, biofilm formation, and virulence, respectively. Additionally, Rli43-regulated target genes were identified using bioinformatic analysis tools and a bacterial dual plasmid reporter system based on *E. coli*. The results showed that the intracellular expression level of the *rli43* gene was significantly upregulated compared with those under extracellular conditions. Compared with the parental and complementation strains, the environmental adaptation, motility, biofilm formation, adhesion, invasion, and intracellular survival of LM-*Δrli43* were significantly reduced, respectively, whereas the LD_50_ of LM-*Δrli43* was significantly elevated in BALB/c mice. Furthermore, the bacterial loads and pathological damages were alleviated, suggesting that sRNA Rli43 was involved in the modulation of the virulence of *L. monocytogenes*. It was confirmed that Rli43 may complementarily pair with the 5′-UTR (−47–−55) of HtrA mRNA, thereby regulating the expression level of HtrA protein at the post-transcriptional level. These findings suggest that Rli43-mediated control was involved in the modulation of environmental adaptation, biofilm formation, and virulence in *L. monocytogenes*.

## 1. Introduction

*Listeria monocytogenes* (*L. monocytogenes*) is a facultative intracellular gram-positive zoonotic pathogen that is ubiquitous bacterium in the natural environment, including soil and sewage [1]. As an important food-borne pathogen, this bacterium has not only raised a serious concern for worldwide public health [2], but also has posed a grave threat to the food industry [3]. Existing studies have shown that the expression of virulence genes can be finely regulated at the transcriptional level by regulatory molecules such as PrfA, Sigma B, and VirR in *L. monocytogenes*, whereby it may adapt to stressful environments rapidly [4].

Recently, a large number of non-coding RNAs (ncRNAs) have been identified in bacteria using bioinformatics, gene chips, and transcriptomic sequencing. Many researchers have revealed that some ncRNAs played important regulatory roles in bacterial adaptation to environmental stress, biofilm formation, motility, quorum sensing, glucose metabolism, and iron homeostasis [5]. As a type of bacterial ncRNAs, small RNA (sRNA) was verified to be involved in the regulation of bacterial virulence and environmental adaptations at the post-transcriptional level through binding to the mRNAs of their target genes [6]. Mraheil et al. found that ncRNA species RliB, Rli33-1, Rli38, and Rli50 are only present in *L. monocytogenes* [7]. Several studies revealed that the absence of rli31, rli33-1, and rli50 could significantly decrease *L. monocytogenes* survival and proliferation in mouse macrophages, whereas the absence of RliB could enhance *L. monocytogenes* proliferation in the mouse liver, which indicated that ncRNA played an important role in regulating *L. monocytogenes* virulence [7,8,9]. Furthermore, Peng et al. confirmed that rli60 may regulate the adaptability of *L. monocytogenes* to environmental stresses, such as low temperature, high temperature, and alkaline and alcoholic conditions [10]. Currently, more than 150 sRNAs have been identified in the *L. monocytogenes* genome [11]; however, the regulatory functions and mechanisms of most sRNAs remain unknown.

Rli43 was identified as an sRNA in *L. monocytogenes* by Toledo-Arana et al. (2009) [8], while its molecular characteristics and regulatory roles in *L. monocytogenes* environmental stress, motility, biofilm formation, and virulence are still unclear. The main purpose of this study was to characterize the intracellular and extracellular expression profiles of sRNA Rli43, to explore its regulatory roles in *L. monocytogenes* environmental stress, motility, biofilm, and virulence, and to identify the target genes regulated by sRNA Rli43. The disclosure of regulatory roles of sRNA Rli43 will provide new insights into the regulatory mechanisms of sRNAs for the environmental adaptation and intracellular parasitism of *L. monocytogenes*.

## 2. Materials and Methods

### 2.1. Primers, Plasmids, Strains, and Culture Conditions

LM EGD-e strain (kindly donated by W. Goebel, University of Wurzburg, Germany) was used as wild-type strain to generate the *Δrli43* mutant and complementation strains. The plasmids pHoss1, pHT304, and pMD19-T were employed for the construction of recombinant vectors [12,13], and pUT18 C and pMR-LacZ were employed for the dual plasmid reporter system [14]. These strains were routinely cultured in BHI liquid medium at 37 °C with vigorous shaking or on agar plates containing 1.5% (wt/vol) agar. For the analysis of bacterial growth, overnight cultures of these strains were diluted 100-fold, inoculated into BHI liquid medium, and cultured under different conditions. The primers were designed using Primer Premier 6.0 software (Premier Canada Assurance Managers Ltd., Laval, QC, Canada) based on the LM EGD-e genome sequence deposited in GenBank (accession number: AL591824) (Table 1). The restriction sites were added to the 5′ end of primers. *E. coli* DH5α was used for plasmid construction, while *E. coli* H101 was used for validation of the interaction between sRNA and target mRNA. All *E. coli* strains were grown in Luria Bertani (LB) solid or broth medium at 37 °C.

### 2.2. Intracellular and Extracellular Expression Profiles of Rli43 Gene

In brief, LM EGD-e strain was inoculated in BHI liquid medium (Sigma, Burlington, MA, USA) and grown to approximately 10^6^ CFU/mL at 37 °C. Following this, mouse RAW264.7 cells cultured in 6-well microplates were infected with LM EGD-e at a macrophage/bacteria ratio of 1:10 for 2 h, and total RNA of LM EGD-e was extracted from the extracellular culture and infected RAW264.7 cells using Trizol (Invitrogen by Life Technologies, Carlsbad, CA, USA), respectively. This was then reversely transcribed into cDNA using an AMV reverse transcription kit (TaKaRa, Shiga, Japan). The *Rli43* gene was amplified using cDNA as template, and *16S rRNA* gene was used as an internal reference gene to profile the differential expression of Rli43 under intra- and extracellular conditions by qRT-PCR.

### 2.3. Generation of Rli43 Gene Deletion and Complementation Strains

The *rli43* deletion mutant and complementation strains were constructed using the homologous recombination technique. Briefly, the *rli43* gene was amplified from the genome of the LM EGD-e strain, and the *Δrli43* mutant gene was generated using the SOE-PCR technique [15]. Following this, the *Δrli43* mutant gene was cloned into the pMD19-T simple vector (TaKaRa, Japan) to generate the recombinant plasmid pMD19-T-*Δrli43*. The *Δrli43* mutant gene was then ligated with pHoss1 to generate the recombinant shuttle plasmid pHoss1-*Δrli43*. pHoss1-*Δrli43* was then electro-transferred (2.5 kv, 5.0 ms) into LM EGD-e competent cells, and these clones were subjected to homologous recombination at 42 °C and erythromycin resistance (10 μg/mL) to obtain *Δrli43* deletion of the mutant strain. For the construction of the complementation strain, the pHT304-*rli43* vector was transferred into LM-*Δrli43* competent cells, and the positive transformant was screened to obtain the complementation strain LM-*Δrli43-rli43.*

### 2.4. Effects of Rli43 Gene Deletion on Responses to Environmental Stresses

The adaptability to environmental stress of LM EGD-e, LM-*Δrli43*, and LM-*Δrli43-rli43* was examined as previously described [16]. In brief, LM EGD-e, LM-*Δrli43*, and LM-*Δrli43-rli43* were inoculated with BHI liquid medium (Sigma, St. Louis, Missouri, USA) and cultured at 30 °C, 37 °C, and 42 °C, respectively, with 200 r/min shaking. The OD_600 nm_ values were measured by full wavelength microplate (Thermo Multiskan SkyHigh, Singapore) at 1.5 h intervals for 12 h, and the bacterial growth curves were plotted. Subsequently, the bacterial solution was inoculated in BHI (Sigma, St. Louis, Missouri, USA) at pH 4 and 9 and containing 3.8% alcohol, respectively, and its OD_600 nm_ value was measured at different times. The growth curve was plotted, and the experiment was repeated 3 times.

### 2.5. Impacts of Rli43 Gene Deletion on Motility and Biofilm Formation

Bacterial motility was determined using semi-solid agar puncture inoculation and a plate diffusion assay as previously described [17]. Briefly, LM EGD-e, LM-*Δrli43*, and LM-*Δrli43-rli43* were punctured and inoculated with 0.5% semi-solid BHI medium for 5 days at 25 °C, respectively, and their growths were observed and photographed daily. Their motility was also assayed by measuring the diameter of the growth zone compared to control bacteria on 0.5% soft agar plates inoculated with 10 μL of a broth culture and incubated overnight at 25 °C. Meanwhile, 200 μL of these strains was added to 96-well microplates containing polystyrene. The formed biofilms were subsequently prepared by crystalline violet staining [17], followed by the assay of OD_570 nm_ values using a spectrophotometer (Shimadzu, Japan) [18]. Following this, the structure of the biofilm was observed under an inverted microscope. Meanwhile, bacteria were cultured on 316 stainless steel sampling plates (Sigma, St. Louis, MO, USA) at 37 °C for 24 h, and observed under a scanning electron microscope (Hitachi SU8010, Japan) for the observation of the morphological structure of biofilm.

### 2.6. Cell Infection Experiments

Briefly, mouse RAW264.7 cells were cultured in 6-well microplates with DMEM supplemented with 10% fetal bovine serum at 37 °C with 5% CO_2_ for 36 h. The cells were infected with LM EGD-e, LM-*Δrli43*, and LM-*Δrli43-rli43* with a macrophage/bacteria ratio of 1:10. Following this, the adhesion, invasion, and intracellular proliferation of RAW264.7 were examined according to the previously described protocol [18,19]. In brief, LM EGD-e, LM-*Δrli43*, and LM-*Δrli43-rli43* were added to the monolayer and incubated at 37 °C with 5% CO_2_ for 1 h. RAW264.7 cell monolayers were washed with DMEM 3 times, followed by 1 h incubation with gentamycin (50 µg/mL) in each well. The monolayers were washed with DMEM thrice. To enumerate the adhered and invading bacterial cells, cell monolayers were subsequently treated with 0.1% Triton X-100, followed by incubation at 37 °C for 10–15 min, and then appropriate dilutions were plated on BHI agar. Plates were incubated at 37 °C for 18–24 h and bacterial cell counts were expressed as percentage adhesion and invasion. Three independent repeats were set up for each group and each experiment was repeated 3 times.

### 2.7. Effects of Rli43 Gene Deletion on Virulence of L. monocytogenes

The experiments with mice were performed following the ethical principles in animal research adopted by the National Council for the Control of Animal Experimentation (CONCEA, Shanghai, China), and the protocol was approved by the Research and Ethical Committee of Shihezi University (No. A2019186). The study was carried out in compliance with the ARRIVE guidelines. All experiments were performed in accordance with relevant guidelines and regulations. At the end of the study, all mice were euthanized by intraperitoneal injection of an overdose of 2, 2, 2-tribromoethanol (500 mg/kg, CAS: 75–80-9 Sigma-Aldrich Chemie GmbH, Steinheim, Germany) and cervical dislocation was performed.

Two hundred 8-week-old BALB/c mice were randomly divided into 1 control group and 3 infection groups. Each group was divided into 5 subgroups with 10 mice each. The mice in each subgroup were injected intraperitoneally with LM EGD-e, LM-*Δrli43*, and LM-*Δrli43-rli43* at 0.5 mL/each, respectively, while the control group was injected with a PBS buffer at 0.5 mL/each. After inoculation, the mental status and mortality of mice in each group were recorded daily for 7 days. The LD_50_ in mice was determined using the Karber method [20], and a Kaplan–Meier survival curve was plotted. For studies measuring bacterial colonization of the liver and spleen, mice were subjected to an intraperitoneal injection of 0.5 mL LM EGD-e, LM-*Δrli43*, and LM-*Δrli43-rli43* at a sublethal dose (5.25 × 10^4^ CFU). At day 1 to day 6 time points post-inoculation, 6 mice in each group were sacrificed by intraperitoneal injection of 2, 2, 2-tribromoethanol (125 mg/kg), which was repeated 3 times. The liver, spleen, and kidney of infected mice were collected and fixed in 4% formaldehyde solution, followed by the preparation of tissue sections. Following this, histopathological changes were observed and recorded.

### 2.8. Determination of the Transcription Levels of Motility, Biofilm, and Virulence-Related Genes

The transcriptional levels of motility-related (*flaA* and *motB*), biofilm-associated (*flgE* and *degU*), and virulence-related genes (*PrfA*, *inlA*, *inlB*, *lap* and *actA*) were determined by qRT-PCR, respectively [21]. Briefly, LM EGD-e, LM-*Δrli43*, and LM-*Δrli43-rli43* were cultured in BHI liquid medium at 25 °C and 37 °C, respectively, which facilitated bacteria to form flagella and biofilm. Total RNA was extracted using Trizol (Invitrogen by Life Technologies, USA) and was reversely transcribed into cDNA using the AMV reverse transcription kit (TaKaRa, Japan). Following this, the relative transcriptional levels of genes mentioned above were analyzed on a LightCycler 480 instrument (Roche, Switzerland), and the 16S rRNA gene was used as an internal reference gene [10,14]. The experiment was repeated 3 times. The relative transcript levels were calculated according to the 2^-ΔΔCT^ method [22].

### 2.9. Analysis of Molecular Characteristics and Target Gene of SRNA Rli43

Genetic localization and molecular characteristics of the *rli43* gene were analyzed using the online software Softberry, fruitfly, and RNAfold, respectively. The phylogenetic tree-based *rli43* gene was then constructed using MEGA10.0 software (NJ method, Bootstrap of 1 000) to reveal the relationships between different strains of *L. monocytogenes*. The genes with low P-values and long consecutive pairings with *rli43* were screened and recognized as target genes by combining TargetRNA2 (http://cs.wellesley.edu/~btjaden/TargetRNA2/) with IntaRNA (http://rna.informatik.uni-freiburg.de/IntaRNA/Input.jsp) (accessed on 21 October 2008) predicting results.

### 2.10. Verification of Interaction between Rli43 and Target Gene Htra Using Dual Plasmid Reporter System

To understand the interaction between the sRNA Rli43 and *htrA* mRNA, the dual plasmid reporter system based on *E. coli* was employed [14]. Briefly, the pUT18C-*rli43* and pMR-LacZ-*htrA* recombinant vectors were constructed by inserting the *rli43* gene and the promoter sequences of target gene *htrA* into pUT18C (Ampicillin resistance) and pMR-LacZ plasmid (Kanamycin and Ampicillin resistance), respectively. The recombinant vectors were co-electrotransferred into *E. coli* BTH101 receptor cells, followed by cultivation in LB liquid medium (containing 100 mg/mL Kanamycin and 100 mg/mL Ampicillin) for the screening of positive clones. The obtained bacteria were further incubated in LB solid medium containing X-gal and IPTG at 37 °C overnight. The color change of the lawn was observed, and the OD_470 nm_ of bacterial solution rinsed from the plates was determined [23]. Furthermore, the mRNA level of the *htrA* gene was determined by qRT-PCR according to the method described above.

### 2.11. Determination of the Expression Level of Target Gene Regulated by Rli43

The protein level of the target gene regulated by Rli43 was detected using SDS-PAGE gels and semi-dry mobile blotting according to the protocols from a previous report [24] with a few changes. Briefly, LM EGD-e, LM-*Δrli43*, and LM-*Δrli43-rli43* protein samples were separated by SDS-PAGE gels and transferred into nitrocellulose membranes (Sigma, St. Louis, MO, USA) using semi-dry mobile blotting (GE Healthcare, Germany). A Western blot was then performed using mouse anti-HtrA recombinant protein antibody as the primary antibody (1: 2000) and goat anti-mouse IgG-HPR (TaKaRa, Japan) (1:4500) as the secondary antibody. Glyceraldehyde-3-phosphate dehydrogenase (GapA) was used as an internal reference protein to analyze the effects of rli43 on the expression of the target protein HtrA. Finally, ImageJ 1.8.0 software (NIH, USA) was applied to quantify the protein bands of the Western blot.

### 2.12. Statistical Analysis of Data

Each experiment was conducted in triplicate, and all data were presented as mean ± standard deviation (SD). The statistical analysis was performed using GraphPad Prism 5.0 software (GraphPad Software, Inc., San Diego, CA, USA). Independent t-tests were used to analyze differences between 2 groups, and a 1-way analysis of variance (ANOVA, London, UK) was used to analyze differences between multiple groups. A value of *p* < 0.05 was considered significant, while *p* < 0.01 was considered extremely significant.

## 3. Results

### 3.1. Expression Profiles of SRNA Rli43 Gene under Intra- and Extracellular Conditions

Given that small RNA (sRNA) is a kind of regulatory molecule that plays an important role in bacteria, the role of sRNA rli43 in the response to environmental changes in *L. monocytogenes* was explored. The sRNA Rli43 was successfully amplified from LM EGD-e by RT-PCR amplification and was verified by sequencing (Appendix A). Following this, we profiled the differential expression of Rli43 under intra- and extracellular conditions by qRT-PCR. The qRT-PCR assay revealed that the transcript level of Rli43 was up-regulated 15.49-fold during cell infection compared with the extracellular culture (Figure 1).

### 3.2. Deletion of Rli43 Gene Reduced Adaptation to Environmental Stress of L. monocytogenes

To investigate the regulatory roles of Rli43 in *L. monocytogenes*, the *rli43* deletion mutant and its complemented strains were constructed by homologous recombination and the overlap-extension PCR technique. The generation of LM-*Δrli43* deletion strain was confirmed by PCR amplification and sequencing verification (Appendix A). Meanwhile, the genetic stability of the LM-*Δrli43* and LM-*Δrli43-rli43* strains were analyzed by the PCR method, respectively (Appendix A). To evaluate the effects of *rli43* gene deletion on adaptability, responses to different environmental stresses of *L. monocytogenes* were determined. As shown in Figure 2, the growth of LM-*Δrli43* significantly declined in the logarithmic phase at 30 °C and 42 °C when compared with LM EGD-e and LM-*Δrli43-rli43* (*p* < 0.05) (Figure 2a,b), while the difference in growth among the three strains was not significant (*p* > 0.05) at 37 °C (Figure 2c). Similarly, LM-*Δrli43* grew significantly lower than LM EGD-e and LM-*Δrli43-rli43* under pH 9 and pH 4 conditions (*p* < 0.05) (Figure 2d,e). However, the difference in growth among the three strains was not significant (*p* > 0.05) under the 3.8% alcohol condition (Figure 2f). Together, these results suggested that deletion of the *rli43* gene could reduce adaptation to environmental stresses of *L. monocytogenes*.

### 3.3. Deletion of Rli43 Gene Weakened L. monocytogenes Motility and Biofilm Formation

To evaluate the motility of LM EGD-e, LM-*Δrli43,* and LM-*Δrli43-rli43*, a semi-solid agar puncture inoculation and a plate diffusion assay were performed. As shown in Figure 3, when grown in semi-solid agar by punctured inoculation, LM-*Δrli43* formed a significantly smaller inverted umbrella structure than that of LM EGD-e and LM-*Δrli43-rli43* (Figure 3a). Furthermore, when grown in swarming agar plates, the motility of the LM-*Δrli43* strain was markedly reduced compared with LM EGD-e and LM-*Δrli43-rli43* (Figure 3b). The movement diameter of LM-*Δrli43* in swarming agar plates was also significantly decreased compared with LM EGD-e and LM-*Δrli43-rli43* (*p* < 0.05) (Figure 3c). To assess biofilm formation, these strains were subsequently grown statically in 96-well microplates containing polystyrene and 316 stainless steel sampling plates in BHI liquid medium at 37 °C for 24 h and 48 h, respectively. The staining of bacterial cells with crystal violet (CV) showed that LM-*Δrli43* formed a looser biofilm structure when compared to LM EGD-e and LM-*Δrli43-rli43* (Figure 4a). We also quantitatively measured biofilm formation ability and the results indicated that biofilm formation in LM-*Δrli43* (OD_570 nm_) was significantly lower than that of LM EGD-e and LM-*Δrli43-rli43* (*p* < 0.05) (Figure 4c). In addition, TEM results indicated that the biofilm formation for LM-*Δrli43* on the stainless-steel surface was also significantly lower than that of LM EGD-e and LM-*Δrli43-rli43* (Figure 4b). The data of the above motility and biofilm formation assays indicates that deletion of the *rli43* gene weakens *L. monocytogenes* motility and biofilm formation.

### 3.4. Deficiency of Rli43 Gene Impaired Adhesion, Invasion, and Intracellular Survival in Macrophage

To probe the regulatory role of Rli43 during the infection of a macrophage, the adhesion, invasion, and intracellular proliferation of RAW264.7 were examined by bacterial cell counts. As shown in Figure 5, the adhesion, invasion, and intracellular survival of LM-*Δrli43* in RAW264.7 cells were significantly lower when compared with LM EGD-e and LM-*Δrli43-rli43* (*p* < 0.05) (Figure 5a–d), respectively, suggesting that *rli43* gene deletion impaired the adhesion, invasion, and intracellular survival in a macrophage.

### 3.5. Deficiency of Rli43 Gene Weakened the Virulence of L. monocytogenes

To investigate the regulatory roles of *rli43* in the virulence of *L. monocytogenes*, we compared the lethality of *L. monocytogenes* in LM EGD-e, LM-*Δrli43*, and LM-*Δrli43-rli43* mice. The mice in each subgroup were injected intraperitoneally with LM EGD-e, LM-*Δrli43*, and LM-*Δrli43-rli43* at 0.5 mL/each, respectively. After inoculation, the mental status and mortality of mice in each group were recorded daily for seven days. As shown in Table 1, the LD_50_ of LM EGD-e, LM-*Δrli43*, and LM-*Δrli43-rli43* on BALB/c mice were 10^5.56^ CFU, 10^7.32^ CFU, and 10^5.76^ CFU, respectively (Appendix A). Compared with LM EGD-e and LM-*Δrli43-rli43*, the LD_50_ of LM-*Δrli43* was elevated by 1.76 and 1.56 logarithmic orders of magnitude, and mice in the LM-*Δrli43*-infected group survived significantly longer (Figure 6a). Moreover, bacterial loads in both liver and spleen were significantly declined (*p* < 0.01) (Figure 6b,c). The pathological changes in the livers, spleens, and kidneys of *L. monocytogenes*-infected mice were further investigated. Compared with the normal control mice (Figure 7a–c), the pathological changes in the livers of LM EGD-e-infected (Black arrows in Figure 7d) and LM-*Δrli43-rli43*-infected (black arrows in Figure 7g) mice were mainly manifested as partial hepatocyte necrosis and hepatic lobular inflammatory cell infiltration; the tissue structure was unclear and contained a large amount of tissue debris. Transparent degeneration and lymphoid tissue necrosis occurred in spleen reticular fibers, and spleen nodules increased in size. Lymphocytes at the site of lymph nodes were necrotic and lymphoid tissue was bleeding (black arrows in Figure 7e,h). Renal venular hemorrhage occurred, inflammatory cells appeared, and the interstitial volume was enlarged (black arrows in Figure 7f,i). However, compared with LM EGD-e-infected and LM-*Δrli43-rli43*-infected mice, histopathological damage to the liver, spleen, and kidney was alleviated to some extent in LM-*Δrli43*-infected mice (Figure 7j–l), indicating that *rli43* gene deletion lessened the virulence of *L. monocytogenes*.

### 3.6. Determination of the Transcription Levels of Motility-, Biofilm-, and Virulence-Related Genes

Based on the results which suggested that the deletion of *rli43* could weaken *L. monocytogenes* motility, biofilm formation, and virulence, we quantified the relative transcription levels of motility-related (*flaA* and *motB*), biofilm-associated (*flgE* and *degU*), and virulence-related genes (*PrfA*, *inlA*, *inlB*, *lap* and *actA*) by qRT-PCR, respectively. As shown in Figure 8, the mRNA levels of motility-related genes *flaA* and *motB* genes (Figure 8a), and biofilm-associated genes *flgE* and *degU* (Figure 8b), were significantly declined in the *rli43* gene deletion strain (*p* < 0.05). Notably, the transcription levels of virulence-related genes (*inlA*, *inlB, lap, actA, PrfA*) were also significantly decreased (*p* < 0.05) in the LM-*Δrli43* strain (Figure 8c). These findings were in agreement with the bacterial phenotype assay, indicating that the deletion of the *rli43* gene can impair the survival and proliferation of *L. monocytogenes* in RAW264.7 cells.

### 3.7. Rli43 Can Modulate the Expression Level of Htra Gene via Targeting Its MRNA

To reveal how Rli43 regulated the phenotypes of motility, biofilm formation, and virulence, we used online software to analyze the genetic localization, molecular characteristics, and target gene of sRNA Rli43. Sequence analysis showed that the *rli43* gene was located at the spacer region between the *inl C* and *rplS* gene in the genome of LM EGD-e, with a length of 254 bp. Moreover, its promoter region contained an FIS transcription factor binding site (Appendix A). It was shown that the transcription of sRNA Rli43 owns six stem-loop domains in the secondary structure (Figure 9a). The genetic evolutionary analysis showed that the *rli43* gene was highly conserved in *L. monocytogenes* serotypes ½ a, ½c, 3c, and 4b (Appendix A). By combining the predicted results of TargetRNA2 and IntaRNA, it was revealed that sRNA Rli43 may complementarily pair with the 5′-UTR (−47–−55) due to the high temperature requirement of htrA mRNA (Figure 9b), suggesting that the *htrA* gene was one of the potential target genes regulated by Rli43. To verify the interaction between rli43 and *htrA* mRNA, a dual plasmid reporter system based on *E. coli* was employed, and the pUT18C-rli43 and pMR-LacZ-*htrA* plasmids were successfully constructed and verified by PCR and double enzyme digestion, respectively (Appendix A). As shown in Figure 9, the lawn of *E. coli* strain co-transformed, as pUT18C-*rli43* and pMR-LacZ-*htrA* was displayed as darker blue on the plates containing X-gal compared with those strains of *E. coli* transformed by pUT18C, pMR-LacZ-*htrA*, or its co-transformation. The OD_470 nm_ values of bacterial suspensions differed significantly (*p* < 0.05) (Figure 9c–g), suggesting that Rli43 can interact with the mRNA of target gene *htrA*. The mRNA and protein levels of the *htrA* gene were significantly lower (*p* < 0.05) in *LM-Δrli43* when compared with LM EGD-e and LM-*Δrli43-rli43* (Figure 9h,i), indicating that Rli43 may modulate the stability of mRNA in the *htrA* gene, so that sRNA Rli43 can positively regulate the expression of the *htrA* gene (Figure 9j).

## 4. Discussion

As a class of ncRNAs, bacterial sRNAs are one of the most important regulators of gene expression and they perform a broad range of physiological functions. Commonly, bacterial sRNAs range from 50–300 nucleotides, and are complementarity with their targeting mRNAs. Currently, a number of sRNAs have been identified in bacteria. However, biological functions have been studied in detail for only a small proportion of sRNAs. Generally speaking, sRNAs are recognized as one of the most important regulators, and they are involved in posttranscriptional or translational control of gene expression through a variety of mechanisms [25]. In *L. monocytogenes*, some sRNAs, together with PrfA, Sigma B, and VirR regulators, form a complex regulatory network to facilitate its survival and infection. As a newly identified sRNA in *L. monocytogenes* [8], however, the regulatory roles and mechanism of Rli43 have not yet been revealed. To investigate the regulatory roles of *rli43* in *L. monocytogenes*, the relative transcription levels of rli43 were profiled under extracellular and intracellular conditions. It was shown that the transcriptional level of Rli43 was significantly upregulated in RAW264.7 cells compared with extracellular conditions, which suggested that Rli43 is involved in the processes of infection and intracellular survival.

An in silico analysis revealed that the high temperature requirement (*htrA*) gene was one of the potential target genes regulated by Rli43. To date, it has been proven that HtrA is an important regulator with a dual function of molecular chaperone and protease activity. It has a critical role in the prevention of severe cellular malfunctions owing to the accumulation of mislocalized or misfolded proteins under physiological and stressful conditions [26]. Currently, HtrA has been proven to be involved in responses to environmental stresses (e.g., high and low temperatures, oxidative stress, high salt, or extreme pH) in *Streptococcus pneumonia* [27], *Lactococcus lactis* [28], *E. coli* [29], and *H. pylori* [30]. Furthermore, Rebecca and Laura et al. confirmed that the survival capacity of the *htrA* gene mutant strains of *L. monocytogenes* is significantly reduced under high temperatures and oxidation conditions [31,32]. Herein, we revealed that Rli43 may facilitate the expression of the *htrA* gene to modulate the adaptation of *L. monocytogenes* in response to environmental changes, which is similar to those bacteria mentioned above.

Several studies have shown that HtrA also played important roles in the regulation of motility and biofilm formation in bacteria [33]. Zhang et al. (2019) confirmed that HtrA is a serine protease from *B. burgdorferi* that regulates the conversion of FlaB [34]. Notably, the motility and the mRNA levels of *flaA* and *motB* genes were significantly reduced in LM-*Δrli43*, suggesting that Rli43 may indirectly regulate motility-related genes through modifying the expression of *htrA* gene. Furthermore, it has been shown that HtrA also plays important roles in biofilm formation [35] in *L. monocytogenes* [10], *Streptococcus pyogenes* [36], *E. coli* [26], and *pneumococcal* [37], respectively. In this study, we verified that the biofilm-forming ability was significantly reduced, and the transcriptional levels of *flgE* and *degU* genes related to biofilm formation were significantly downregulated in the deletion strain. However, the detailed mechanism of LM Rli43-modulating motility and biofilm formation should be further unraveled in the future.

It was confirmed that HtrA also played important regulatory roles in the expression of bacterial virulence genes [26,27,38,39,40,41,42,43], thereby influencing bacterial pathogenicity [31]. In *Campylobacter jejuni*, it was confirmed that HtrA could modify the processes of cell adhesion, invasion, and migration ability [44]. In particular, the deficiency of the *htrA* gene could lessen the survival capacities of *Salmonella*, *Brucella,* and *Yersinia* in macrophages or mice [45]. In *L. monocytogenes*, *htrA* gene deletion may result in the accumulation of misfolded proteins on the bacterial membrane, thus impairing intracellular survival [46,47]. In the present study, we verified that the deficiency of the *rli43* gene impeded adhesion, invasion, and intracellular survival in a macrophage. In addition, the qRT-PCR results further verified that these findings were in agreement with the results of the bacterial phenotype assay, indicating that the deletion of the *rli43* gene can impair the survival of *L. monocytogenes* in RAW264.7 cells.

Current studies have revealed that the molecular mechanisms underlying sRNA-mediated control are obvious diversity and flexibility in bacteria [3,6]. Commonly, sRNAs may regulate the expression of target genes by base pairing them with the 5′-untranslated region (5′-UTR) of the mRNA, thereby modulating multiple physiological processes, such as virulence and biofilm formation of bacteria [48]. On the one hand, the binding of trans-encoded sRNAs in the vicinity of the ribosomal binding site (RBS) may result in the inhibition of translation initiation or mRNA decay. Alternatively, sRNAs are likely to promote translation or prevent mRNA degradation, especially by base pairing to the far upstream from the RBS [6]. In this study, we verify the interaction between rli43 and target gene htrA mRNA using a two-plasmid system based on *E. coli*, which suggested that rli43 may bind to the 5′-UTR of htrA mRNA. To date, at least five RNases have been identified in bacteria, namely RNase A, RNase P, RNase E, RNase R, and RNase III. Among them, RNase III is mainly involved in the degradation of double-stranded RNA, while RNase JI is mainly involved in the degradation of single-stranded RNA in gram-positive bacteria. Combined with the results of the qRT-PCR and the Western blot, it is reasonable to infer that Rli43 may protect htrA mRNA from RNase JI degradation, thereby maintaining the stability of its mRNA. This regulatory mechanism mediated by rli43 is similar to the regulatory mode reported by Dutta et al. [48], which further highlights the similarities of sRNA-mediated control in various species of bacteria. However, this detailed regulatory motif in Rli43 should be further experimentally validated by employing site-directed mutagenesis.

## 5. Conclusions

This study revealed that Rli43 was involved in posttranscriptional control of *htrA* gene expression by modulating the stability of its mRNA. The regulatory mechanisms of sRNA rli43 provided new insights into the understanding of the diversity and flexibility of sRNA-mediated control in biofilm formation and virulence in *L. monocytogenes*.

## Figures and Tables

**Figure 1 pathogens-11-01137-f001:**
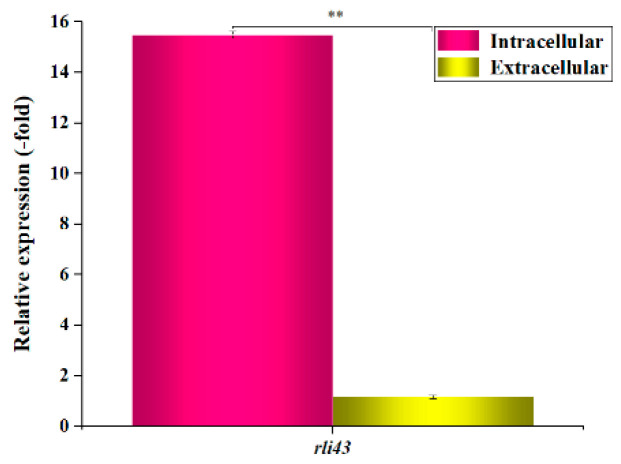
Expression profiles of sRNA *rli43* gene under intra- and extracellular conditions. *16S rRNA* gene was used as an internal reference gene, error bars represent the standard deviation of three biological replicates (** *p*-value  <  0.01).

**Figure 2 pathogens-11-01137-f002:**
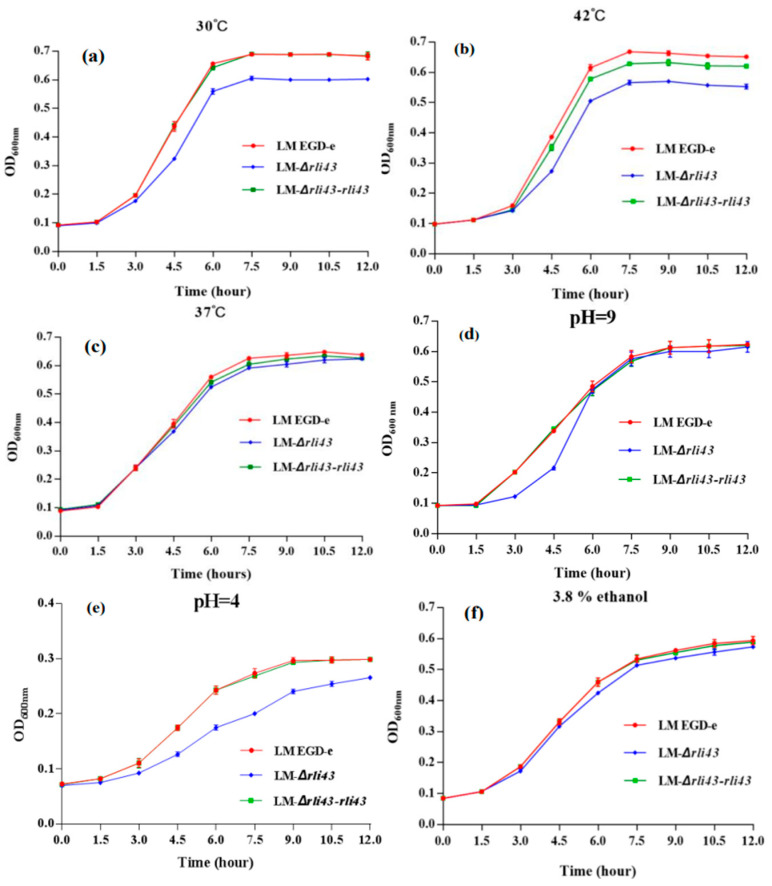
Growth curves of LM EGD-e, LM-*Δrli43*, and LM-*Δrli43-rli43* under different stress conditions. (**a**–**f**) growth curves of LM EGD-e, LM-*Δrli43*, and LM-*Δrli43-rli43* at 30 °C, 42 °C, 37 °C, pH 9, pH 4, and 3.8% ethanol, respectively. All data are expressed as the means ± SD (n = 3). The differences between strains were assessed by one-way ANOVA.

**Figure 3 pathogens-11-01137-f003:**
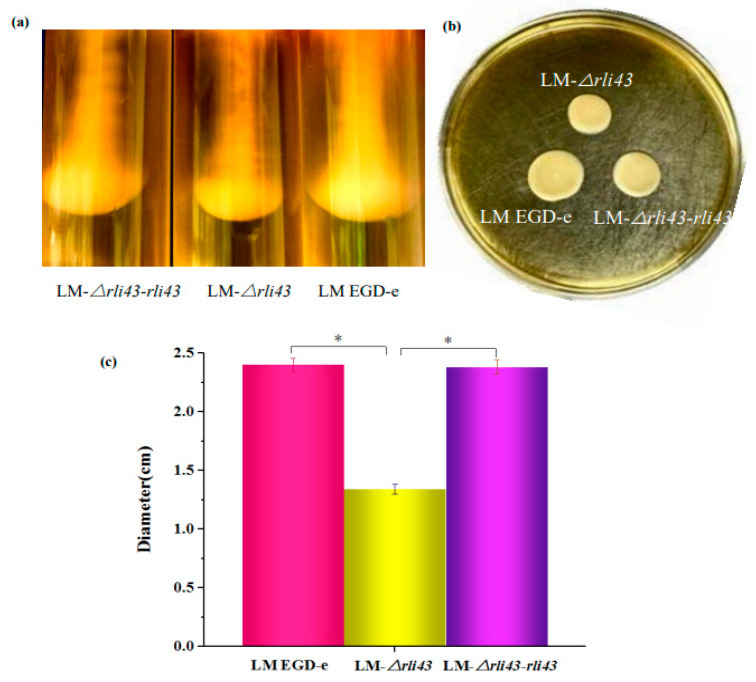
Comparisons of motility of different strains of *L. monocytogenes*. (**a**) puncture inoculation of different strains of on semi-solid medium; (**b**) swarming motility of different strains on 0.5% soft agar plates; and (**c**) comparison of the diameters of swarming motility of different strains on 0.5% soft agar plates. Data are the mean ± SD (*n* = 3). The differences between strains were assessed by one-way ANOVA (* *p*-value  <  0.05).

**Figure 4 pathogens-11-01137-f004:**
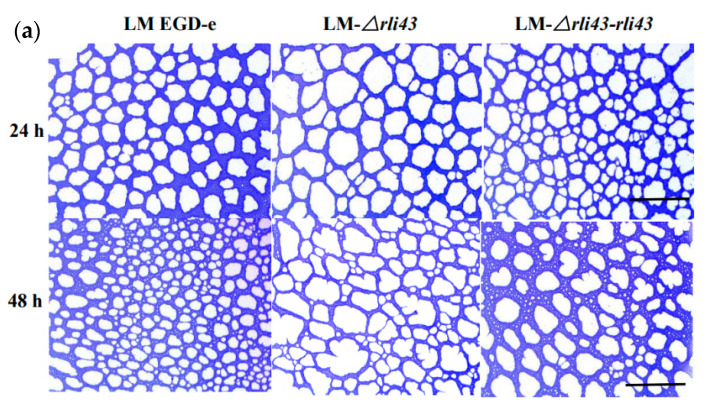
Determination of capacities of biofilm formation of different strains of *L. monocytogenes*. (**a**) biofilm formation on polystyrene surfaces observed by microscopic (20 × 10), scale bars, 0.4 mm; (**b**) biofilm formation on stainless steel coupons observed by scanning electron micrographs (×10,000), scale bars, 20 μm; and (**c**) assay of biofilm formation ability using OD_570 nm_. The differences between strains were tested by one-way ANOVA (* *p*-value  <  0.05). Data represent the mean ± SD (*n* = 3).

**Figure 5 pathogens-11-01137-f005:**
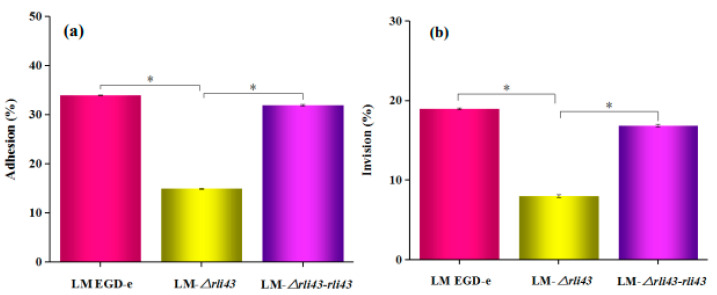
Determination of cell adhesion and invasion capacity of different strains of *L. monocytogenes*. (**a**) Adhesion rate of LM EGD-e, LM-*Δrli43*, and LM-*Δrli43-rli43* in RAW264.7 cells. (**b**) Invasion rate of LM EGD-e, LM-*Δrli43*, and LM-*Δrli43-rli43* in RAW264.7. (**c**) Bacterial numbers in RAW264.7 cells infected by LM EGD-e, LM-*Δrli43*, and LM-*Δrli43-rli43* at various periods. Data represent the mean ± SD (*n* = 3). Asterisks denote statistically significant difference by one-way ANOVA between strains (α = 0.05) with *p* < 0.01 and *p* < 0.05. (**d**) Inhibition rate of LM EGD-e, LM-*Δrli43*, and LM-*Δrli43-rli43* in RAW264.7 cells after infection at time points. All data are expressed as the means ± SD (*n* = 3). The differences between strains were assessed by one-way ANOVA (* *p*-value  <  0.05, ** *p*-value  <  0.01).

**Figure 6 pathogens-11-01137-f006:**
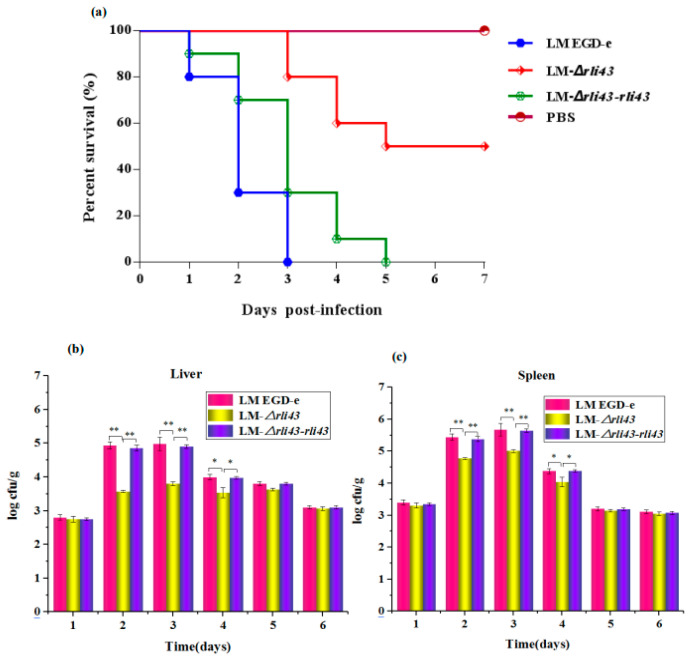
Survival curves and bacterial loads in mice infected with different strains of *L. monocytgenes*. (**a**) survival curves; (**b**) bacterial loads in livers; and (**c**) bacterial loads in spleens. Results are represented as means ± SD of three experiments and expressed as log CFU/g. The differences between strains were assessed by one-way ANOVA (* *p*-value  <  0.05, ** *p*-value  <  0.01).

**Figure 7 pathogens-11-01137-f007:**
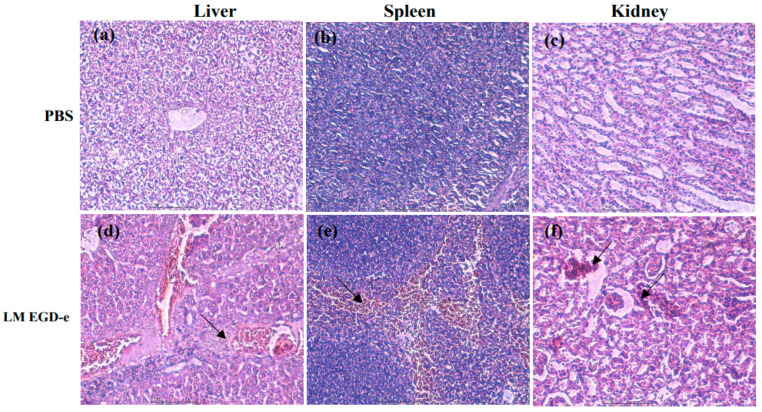
Histopathological examinations of the liver, spleen, and kidney of mice infected with LM (HE × 400, scale bars, 100 μm). (**a**–**c**) Liver, spleen, and kidney from mice injected with PBS. (**d**–**f**) Liver, spleen, and kidney from mice infected with LM EGD-e. (**g**–**i**) Liver, spleen, and kidney from mice infected with LM-*Δrli43-rli43*. (**j**–**l**) Liver, spleen, and kidney from mice infected with LM-*Δrli43*. Arrows indicate major lesion areas.

**Figure 8 pathogens-11-01137-f008:**
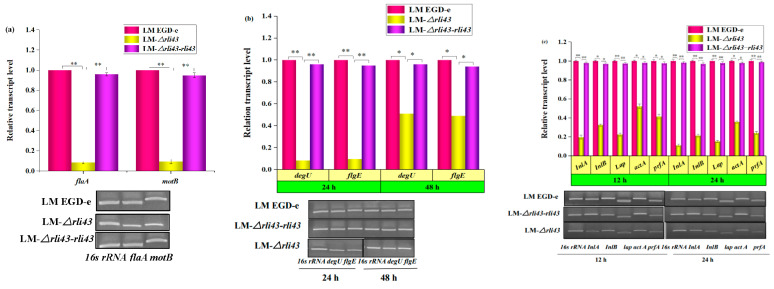
Analysis of the relative transcriptional levels of motility-, biofilm-, and virulence-related genes. (**a**) Relative transcriptional levels and qRT-PCR electrophoretic band analysis of *flaA* and *motB* gene of LM EGD-e, LM-*Δrli43*, and LM-*Δrli43-rli43*, respectively. Asterisks denote significance in gene transcription compared with different strains according to one-way ANOVA (** *p*-value  <  0.01). (**b**) Relative transcriptional levels and qRT-PCR electrophoretic band analysis of *degU* and *flgE* gene of LM EGD-e, LM-*Δrli43*, and LM-*Δrli43-rli43*, respectively. The differences between strains were tested by one-way ANOVA (* *p*-value  <  0.05, ** *p*-value  <  0.01). (**c**) Relative transcriptional levels and qRT-PCR electrophoretic band analysis of *InlA*, *InlB, Lap, actA*, and *prfA* gene of LM EGD-e, LM-*Δrli43*, and LM-*Δrli43-rli43* at 12 h and 24 h, respectively. All data are expressed as mean ± SD (*n* = 3). Asterisks denote significance in gene transcription compared with different strains via one-way ANOVA (α = 0.05, * *p*-value  <  0.05, ** *p*-value  <  0.01).

**Figure 9 pathogens-11-01137-f009:**
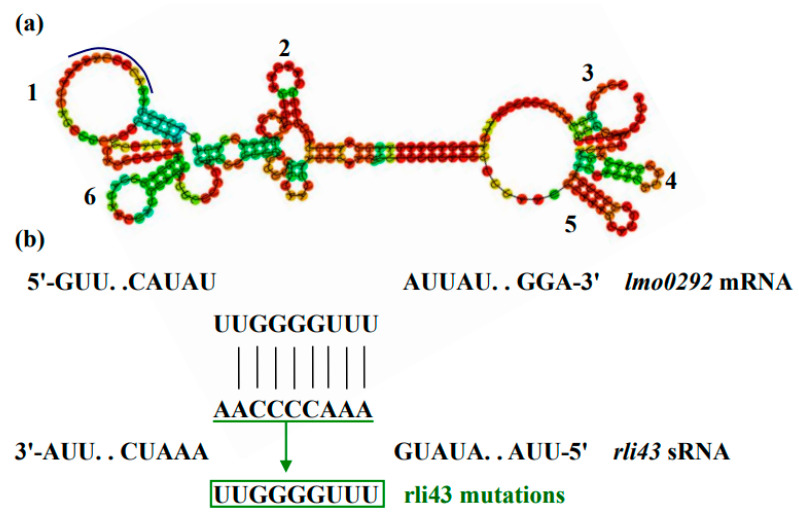
Verification of the interaction between sRNA Rli43 and target gene *htrA* using dual plasmid reporter system. (**a**) Secondary structure of sRNA Rli43 (six stem-loop structures, the blue line represents base-paring region). (**b**) The predicted potential gene targeted by Rli43, and the nucleotide mutations in the green box for the sRNAs. (**c**) Bacterial lawn of BTH101 co-transformed by pUT18C and pMR-LacZ. (**d**) Bacterial lawn of BTH101 co-transformed by pUT18C and pMR-LacZ- *htrA*. (**e**) Bacterial lawn of BTH101 co-transformed by pUT18C-*Δrli43* and pMR-LacZ- *htrA*. (**f**) Bacterial lawn of BTH101 co-transformed by pUT18C-*rli43* and pMR-LacZ-*htrA*. (**g**) Bacterial suspension determined by OD_470 nm_. (**h**) Determination of the mRNA level of *htrA* genes by qRT-PCR. (**i**) Determination of the expression of HtrA protein by Western blot. All data represent the mean ± SD (*n* = 3). The differences between strains were tested by one-way ANOVA (* *p*-value  <  0.05, ** *p*-value  <  0.01). (**j**) The mechanism of sRNA Rli43 to modulate the mRNA level of *htrA* gene by targeting its 5′-UTR region.

**Table 1 pathogens-11-01137-t001:** Primers used in the study.

Primer Names	Primer Sequences (5′→3′)	
rli43 F	TTATTTGCAATTTTTCTCAATAAGTA	RT-PCR/qRT-PCR
rli43 R	ACCATCCCGCGCCATTTA	
R1	CGG*GGTACC*CACAAAAGAGGATTTTCATATACTTCC	SOE-PCR
R2	GCAGAAAGTTGGGTATGACTATAGAAAAAAAGCCTTGTCA	
R3	TGACAAGGCTTTTTTTCTATAGTCATACCCAACTTTCTGC	
R4	AA*AACTGC*AGCCTACGTGGTAAAGCGGCTC	
P1	CCC*AAGCTT*TTATTTGCAATTTTTCTCAATAAGT	Complementation strains
P2	G*GAATTC*TAAATGGCGCGGGATGGT	
16S rRNA F	GAGCTAATCCCATAAAACTATTCTCA	qRT-PCR
16S rRNA R	ACCTTGTTACGACTTCACCCC	
flaA F	AACAAGCAACTGAAGCTATTGATGAATT	
flaA R	TGCGGTGTTTGGTTTGCTTGA	
motB F	AATCGCCAAAGAAATCGGCG	
motB R	CGCCGGGGTTTACTTCACTA	
flgE F	AATGCCAACACGACAGGATA	
flgE R	TTTGTTCCAGCGTAAAGTCC	
degU F	GAGGTCGTAGCGGAAGCTGA	
degU R	CTGTTACATATTCATCTGTATCA	
InlB F	CTGAAAAAAATGGTGGGCATGAG	
InlB R	CCGCCATTTCGGGCTTCTCT	
PrfA F	ACGGGAAGCTTGGCTCTATT	
PrfA R	TGCGATGCCACTTGAATATC	
Lap F	GCGGCCCAAGAAGTATTAGAA	
Lap R	CTGTCGCGAAGATGTTTTTAATACA	
InlA F	TGTGACTGGCGCTTTAATTG	
InlA R	TCCAATAGTGACAGGTTGGCTA	
actA F	ATGCGGGGAAATGGGTACGT	
actA R	CCAAGAAGCATTGGTGTCTCTGG	
P3	GGATCCAACATTCTCCACTCCTTAAAAA	Dual plasmid reporter system
P4	GGTACCATCATTTAACGGCTGAAACCCT	
P5	CCAAGCTTGCTTTTTCATCGTTTTTTCA	
P6	GGGGTACCATGTCTAGCAGGTTCTGATTCT	
P7	*GGATCC*CATTCTCCACTCCTTAAAAATA	
P8	AACTACATATATGTTTGGGGTTAAATCAGCTGCC	
P9	GGCAGCTGATTTAACCCCAAACATATATGTAGTT	
P10	GG*GGTACC*CAAGGGTTTCAGCCGTTAAATGA	
P11	ATGGACGAGAAAGAAAAGAATT	
P12	AATAACACCAATAAGTGCTGTT	

## Data Availability

All data generated or analyzed during this study are included in the manuscript/Appendix A. All study original images are included in the manuscript (supplementary information files: Original Images for Blots-Gels file). The sequencing data are available in the GenBank repository, and the GenBank accession number for LM EGD-e genome sequence is AL591824 (https://www.ncbi.nlm.nih.gov/nuccore/AL591824 (accessed on 27 February 2015)). The sequence of *rli43* gene is available in the GenBank repository (https://www.ncbi.nlm.nih.gov/genbank/ (accessed on 6 March 2022)).

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
