# Peer review of "A Regulatory SRNA Rli43 Is Involved in the Modulation of Biofilm Formation and Virulence in Listeria monocytogenes"

_pathogens, 2022, doi:10.3390/pathogens11101137_

Round 1
Reviewer 1 Report
This is a very interesting study, which seems reasonably designed. However, important information is missing to allow others to repeat the experiments or to fully judge the experimental design. So, as my only major comment, this information has to be added.
detailed comments of major importance
l67: "LM EGD-e strain": please add a source. Is this a strain from an official collection like ATCC, DSMZ? Then provide a number. If not, provide a reason, why you did not use such a strain. (partially clarified later on, but still: provide a reason for the choice of this strain)
l70: "These strains": there is only one strain mentioned yet. Do you mean its transformed variants?
l72: I miss the culture conditions plus timepoints of analysis. Might come late, but then change the heading.
l93: okay, but should be added to the first mention of the strain, not here. Describe the reason to use exactly that strain at the beginning of the methods section
l100 "at 42 °C and erythromycin resistance (10 μg/mL) ": unclear, rephrase. Please make clearer who you could differentiate (confirm) the wild-strains, the deletion strain and the re-complemented strain in the end.
l106: how many cfu were inoculated and how did you check that inoculum?
l108: Which instrument was used to measure OD600 nm? Did you assess OD in plates or in cuvettes? How did you validate these meaurements? How many replicates of the control were performed?
l120: which control bacteria?
l124: add a reference for assessing turbidity at OD570 nm, not OD600. Where was OD assessed, in plates or cuvettes? Means: in the sessile cells or in the suspension?
l137: did you perform a gentamicin control without cells, to assess the bias due to spontaneous mutation? If not: should be done!
l177: please add a reference and reason for using the 16SrRNA gene as an internal control in the Delta-Delta-Ct method. To my knowledge, there are some criticisms about using that gene. How many 16S copies are present in your strain? Did you correct for copy number?
l183 "relationship between different strains": which strains, which source? Sequence data from Genbank? Which?
l195: please add which plasmid is marked by which of these two resistances
l198: now, I assume that this was done in cuvettes. This is the third different OD for measuring bacterial suspensions now, please add information on the reasons for choosing them differently (e.g. validation experiments or references recommending that).
l198: Bacterial solution rinsed with what liquid medium?
Figure 1 y-axis: unit is "-fold", I think? Please clarify.
Figure 2: experiments wee performed in three replicates, but only one point is shown. a) Why did you choose the mean, not the median? b) please add information on variance, e.g. as an error bar, which might show the standard deviation, if values follow normal distribution (would be hard to proof with 3 values only), or might simply represent min and max values, if not.
Figure 7: most bacteriologists are not that familiar with pathology, so please add an extended figure caption and arrows that show, what should be seen in the different parts of the figure
Figure 8, figure caption: please add information on what is seen in the lower part of the figures (western blot). How do you explain that you see significant differences on the transcript level, while the semiquantitative information obtained from the western blot (thickness of bands) does not seem to reveal any difference, at least in Figure 8a?
Figure 9f: -rli43: unclear, is that Deltarli43-rli43, as termed before? Then I would prefer that term, since otherwise it is hard to capture the difference on a first glance, and -rli43 could also be read as "minus rli43", but it is re-complemented, instead, to my understanding?
Discussion: please extend, some parts of the results section are hardly discussed yet. Please also strengthen the conclusions, you have nice and clear results, so "provided new insights" is phrased unnecessarily cowardly.
Minor comments:
Seek help with language editing, there is a fault in the very first sentence of the abstract.
l31: Rephrase "intracellular parasitic" to "facultative intracellular"
l34: "but also has posed a grave threat to the livestock industry": this is not true, there are no regulations or programs to defeat listeria in livestock, so there are no consequences at all, except in case of mastitis. "Food industry" would fit better, but the literature source might have to be changed
l36: the genes or not "virulent", but code for virulence, please change to "virulence genes"
l77/78: italics missing
l229 "compared to extracellular culture, which was up-regulated in ...infected macrophages" - unclear, wrong relation, I think.
Reviewer 2 Report
This manuscript by Wang et al. examines the function of the non-coding RNA rli43 in Listeria monocytogenes (Lm). The study is premised on the observation that rli43 transcript is increased in during intracellular growth of Lm. The authors then generate a genetic deletion of rli43 and examine how the RNA impacts bacterial behavior. Loss of rli43 reportedly reduces growth in stress conditions, bacterial motility, and biofilm formation. Rli43 is also required for full virulence in both in vitro and in vivo pathogenesis models. Deletion of rli43 also reduces the transcription of several motility-, biofilm-, and virulence-related genes. Using bioinformatic tools, the authors identify the 5’-UTR of htrA, a cell surface chaperone, as a potential binding partner for rli43. Using a bacterial 2 plasmid system, they report that rli43 can modulate the expression of a gene which contains the 5’-UTR from htrA.
Bacterial small RNAs have been an understudied component of the regulatory networks that control bacterial physiology and virulence. This study represents a potentially important addition to the literature as the authors take a previously identified small RNA and demonstrate its importance in a variety of bacterial processes. The generation of an rli43 genetic deletion provides convincing evidence of its connection to several phenotypes. While the authors’ identification of htrA as a potential target of rli43 regulation is intriguing, no direct evidence is presented that this mechanism operates in Lm. The proposed mechanism also cannot account for the observed transcriptomic changes in Δrli43. Finally, the manuscript would benefit from additional editing and further description and referencing of the methods used.
Specific comments:
1. Materials and methods – A number of sections of the materials and methods lack sufficient detail to be reproducible. If the authors intend to reference previously published methods or manufacturers’ protocols, they should explicitly do so in the text. Some specific examples include:
Section 2.1 – References should be provided for the plasmids where available.
Section 2.2 – How many RAW cells were used? How many bacteria? How long was the incubation period? What primer and conditions were used to produce cDNA? How was the qPCR performed and quantified?
Section 2.3 – Which primers were used for the SOE PCR?
Section 2.6 – How many bacteria and cells were used for infection experiments? A MOI of 1:100 cells to bacteria is extremely high for macrophages. A lower MOI of 1:1 or even 1:10 should be tested for comparison. Methods for determining adherence vs internalization should be more clearly described.
Section 2.7- How were tissue specimens prepared and stained prior to microscopy?
Section 2.10 – How were the recombinant plasmids in this experiment produced? What primers were used for qRT-PCR?
Section 2.11 – What was the source of the antibodies used for western blotting?
2. Figure 1. How do the bacterial numbers (CFU) compare between the extracellular and intracellular experiments? I realize that the results are normalized, but it would be helpful to know how the actual bacterial numbers compare.
3. Figure 6b-c. Was statistical analysis performed on log-transformed CFU counts or on raw data? The number of bacteria recovered from infected mice often varies over several orders of magnitude and is not normally distributed. An ANOVA requires data which is normally distributed, and a violation of normality can result in overestimation of statistical significance. We recommend either performing statistical analysis on log-transformed data, which typically is normally distributed, or utilizing a non-parametric test (Kruskal-Wallis test) which does not require normality. Also, it is best to indicate individual mice with data points.
4. Figure 8. The authors identify a similar decrease in transcript level for all tested genes even though they are part of multiple functional groups and not proposed to be directly regulated by rli43. This raises a concern that the finding may be an artifact or caused by non-specific changes after deletion of rli43. Did the authors identify other genes that were not altered in Δrli43? Did they test other genes not expected to be regulated by rli43?
5. Figure 9c-g. The authors utilize an unusual procedure of growing lacZ expressing bacteria as a lawn with substrate, washing them off the plate, and measuring absorbance of the suspension. Is there a reason the authors chose this approach rather than a more traditional Miller assay[1]? Is there support for it being a reliable measure of beta-galactosidase expression? How does the copy number of the plasmids compare? Could copy number difference impact results?
Minor comments:
1. Line 36: ‘virulence genes’ not ‘virulent genes’.
2. Line 268: were the plates made of polystyrene or did they actually contain bits of polystyrene?
3. Figure 5d: what is meant by ‘inhibition rate’ in the figure legend? This should be clarified.
[1] See e.g. Zhang et al. (1995) Control of the Escherichia coli rrnB P1 Promoter Strength by ppGpp*. JBC 270(19): 11181-11189.
Round 2
Reviewer 2 Report
The authors have satisfactorily addressed earlier noted concerns.